# Targetoid Skin Lesions in a Child: Acute Hemorrhagic Oedema of Infancy and Its Differential Diagnosis

**DOI:** 10.3390/ijerph16050823

**Published:** 2019-03-06

**Authors:** Francesco Miconi, Lorenzo Cassiani, Emanuela Savarese, Federica Celi, Manuela Papini, Nicola Principi, Susanna Esposito

**Affiliations:** 1Pediatric Clinic, Department of Surgical and Biomedical Sciences, Università degli Studi di Perugia, 06129 Perugia, Italy; Francesco.miconi90@gmail.com (F.M.); manu-S84@hotmail.it (E.S.); 2Dermatologic Section, Università degli Studi di Perugia, 05100 Terni, Italy; lorenzocassiani08@gmail.com (L.C.); Manuela.papini@unipg.it (M.P.); 3Pediatric Clinic, Azienda Ospedaliera di Terni, 05100 Terni, Italy; federicaceli@gmail.com; 4Università degli Studi di Milano, 20122 Milan, Italy; nicola.principi@unimi.it

**Keywords:** acute hemorrhagic oedema of infancy, Finkelstein disease, pediatric dermatology, Seidlmayer disease, vasculitis

## Abstract

Acute hemorrhagic oedema of infancy (AHEI) is a cutaneous leukocytoclastic small-vessel vasculitis presenting with localized purpuric large skin plaques that are frequently associated with fever and oedema. It must be promptly differentiated from a number of diseases with similar dermatologic manifestations with potentially severe clinical courses that require adequate monitoring and prompt therapy to avoid the risk of a negative evolution. A 15-month-old girl with a negative personal medical clinical history was admitted for the sudden appearance of petechiae on the soft palate. The patient was moderately febrile during the following two days, with a maximum ear temperature of 38.3 °C. The fever disappeared on the third day, whereas the hemorrhagic rash progressively increased and extended to the limbs, face and auricles associated with a strong oedematous component. Moreover, on the second day of hospitalization, bilateral oedema of the metacarpophalangeal joints with joint pain appeared. The blood and serological tests showed an increase in C-reactive protein concentration (3.58 mg/dL) in the absence of leukocytosis and with a normal platelet count (180,000/mm^3^). The examination of the peripheral smear showed the presence of some large mononuclear elements with hyperbasophile cytoplasm. No alterations in platelet morphology were evidenced. The skin manifestations progressively diminished and disappeared spontaneously within 3 weeks, leaving no sequelae. *Conclusion*: This case shows the classic skin lesions of AHEI that require differentiation from those of more severe diseases that need prompt recognition and therapy. In this case, the age of the patient, the lack of systemic involvement and the favorable clinical course without therapy were typical. However, as these patients may present to the emergency department with an impressive clinical picture, the condition must be promptly diagnosed to avoid unnecessary diagnostic procedures and to reassure parents.

## 1. Introduction

Acute hemorrhagic oedema of infancy (AHEI), also named Finkelstein disease, Seidlmayer disease, Finkelstein-Seidlmayer disease, rosette form purpura, infantile post-infectious iris-like purpura and medallion-like purpura, is a cutaneous leukocytoclastic small-vessel vasculitis presenting with localized purpuric large skin plaques frequently associated with fever and oedema [1]. In recent years, evaluation of described cases, including analysis of skin biopsies, has led researchers to establish that AHEI is a mild, benign disease with spontaneous resolution within a few weeks. However, it must be promptly differentiated from a number of diseases with similar dermatologic manifestations with potentially severe clinical courses that require adequate monitoring and prompt therapy to avoid the risk of a negative evolution [2]. Here, a case of AHEI is presented, and clinical findings that can differentiate AHEI from common mimics are discussed.

## 2. Case Presentation

A 15-month-old white Caucasian girl with a negative personal medical clinical history was seen by her family pediatrician for the appearance of petechiae on the soft palate during the last 24 h. As she had been vaccinated with the measles-mumps-rubella (MMR) vaccine 12 days before and had presented a mild episode of febrile diarrhea 8 days beforehand, immune thrombocytopenic purpura (ITP) was suspected. However, a platelet count was immediately performed and excluded this diagnosis (171,000 platelets/mmc^3^). No drug was prescribed, and the decision was made to wait and see the evolution of the disease. The patient was moderately febrile during the following two days, with a maximum ear temperature of 38.3 °C. The fever disappeared on the third day, whereas the hemorrhagic rash progressively increased and extended to the skin in different parts of the body. The diameter of hemorrhagic lesions varied from few millimeters to several centimeters (Figure 1).

Hospitalization was decided. At admission, the patient’s general condition was good, but a rash characterized by petechial-hemorrhagic lesions with sharp merges of varying sizes localized to the limbs, face and auricles associated with a strong oedematous component was evidenced (Figure 2).

Moreover, on the second day of hospitalization, bilateral oedema of the metacarpophalangeal joints with joint pain appeared. The articular manifestations were responsive to analgesic therapy with paracetamol and resolved spontaneously in a week.

Blood counts, hepatic and renal function, C-reactive protein, coagulation, antineutrophil cytoplasmic autoantibodies, serology for Epstein Barr virus, cytomegalovirus, Rubella, Herpesvirus, parvovirus B19, fractions C3 and C4 of the complement, pharyngeal swab, blood culture, urinalysis, factor V leiden, D-dimer, erythrocyte sedimentation rate and peripheral blood smear were collected. The blood and serological tests showed an increase in C-reactive protein concentration (3.58 mg/dL) in the absence of leukocytosis and with a normal platelet count (180,000/mm^3^). The examination of the peripheral smear showed the presence of some large mononuclear elements with hyperbasophile cytoplasm. No alterations in platelet morphology were evidenced. Ultrasound examination of the abdomen and the urinary tract excluded visceral involvement. To avoid an invasive procedure in a child with an already well-defined diagnosis, cutaneous biopsy was avoided, given the good general condition of the patient.

The skin manifestations progressively diminished and disappeared spontaneously within 3 weeks, leaving no sequelae.

Management of the case was approved by the Ethics Committee of Santa Maria Hospital, Terni, Italy (2018-PED-01). The patient’s parents provided their written informed consent for the management of their child and the publication of the case report (including the Figures).

## 3. Discussion

AHEI is a rare disease. In 2013, it was reported that approximately 300 cases had been described since the first report discussed the disease approximately a century ago [3,4]. AHEI occurs predominantly in young children; approximately 80% of the cases have been described in patients aged 6 to 24 months [3]. In a great number of cases, skin lesions are preceded by a prodrome, usually a mild respiratory infection. This progression has suggested a potential role of bacteria or viruses as triggers of this vasculitis. Emergence of AHEI after *Streptococcus pneumoniae* [5], *Mycoplasma pneumoniae* [6], rotavirus [7], and coxsackie virus infection [8] has been reported. In some patients, a temporal association with vaccine administration has been described. AHEI was reported after vaccination with BCG, influenza, *Hemophilus influenzae* type B, diphtheria, tetanus, acellular pertussis, hepatitis B, polio, and conjugate pneumococcal vaccines [3,9,10,11,12]. In at least two cases, as in the one described here, previous immunization with MMR vaccine has been reported, [13,14]. At presentation, more than 90% of described children have a nontoxic appearance, mild fever is present in approximately 50% of the cases, and in a similar percentage of children, the white blood cell count, erythrocyte sedimentation rate, and C-reactive protein serum levels are slightly increased. However, clotting studies and complement protein levels are in the normal range, autoantibodies are absent, and urinalysis is usually normal [1]. Skin lesions are nonpruritic and are represented by large red to purpuric plaques mainly located on the legs, arms, face and ears, whereas the trunk is relatively unaffected. Nonpitting, often indurative oedema in the same sites, particularly in the distal extremities, is frequently associated. Lesions of the mucous membranes are rare. In some cases, skin lesions are very impressive and appear in contrast to the patients’ good general conditions. Clinical evidence for abdominal involvement is absent. Articular pain in the sites of the oedema is reported in 20% of the cases. In the patient here reported, the age of the patient and most of the signs and symptoms including articular pain, were those characteristically reported in children with AHEI. After exclusion of ITP, the diagnosis was relatively easy, as it was the differentiation of AHEI from the other clinical conditions with cutaneous hemorrhagic manifestations that were or weren’t associated with urticaria that may be confused with AHEI. Among them, Henoch-Schönlein purpura (HSP), meningococcaemia, Kawasaki disease, erythema multiforme are the most common forms. These diseases need accurate diagnosis and, at least some of them, prompt therapy to avoid the risk of a negative rapid evolution. Contrarily, as in this case, AHEI evolution is always benign, and skin manifestations disappear in approximately 3 weeks in 80% of the cases [1,2]. Drug treatment is not needed, as corticosteroids, nonsteroidal anti-inflammatory drugs, and antihistamines do not reduce clinical manifestations and do not alter the course of the disease.

HSP is an acute immunoglobulin A-mediated disorder characterized by a generalized vasculitis involving the small vessels of the skin, the kidneys, the gastrointestinal tract, the joints, and, sometimes, the lungs and the central nervous system (CNS). It differs from AHEI by the sites of both the purpura and the oedema: the purpura has a buttock and lower extremity predominance and does not occur on the face and ears; the oedema is periorbital and peripheral. Moreover, the rash tends to persist for a longer time (>1 month). The age of the patients (3–8 years) is also different. Furthermore, systemic findings, such as abdominal pain, renal disease, and arthritis, are common and can negatively influence final prognosis of the disease [15].

Meningococcemia is the consequence of the dissemination of *Neisseria meningitidis* into the bloodstream and may present with a purpuric rash. However, the rash is constantly accompanied by systemic signs, toxic appearance, lethargy and poor feeding and, in cases of meningitis, by signs and symptoms of CNS involvement, absent in cases of AHEI [16].

Kawasaki disease is an acute vasculitis that occurs mainly in children aged 18–24 months [12]. It is characterized by high fever, a polymorphic rash, changes in the lips and oropharyngeal mucosa, bilateral, nonexudative, bulbar conjunctival injection and unilateral, non-suppurative cervical lymphadenopathy. Moreover, swollen hands and feet are commonly evidenced [17].

Erythema multiforme is an acute, self-limited, and often recurring skin condition that is considered to be a type IV hypersensitivity reaction associated with certain medications, infections, and other various triggers, such as vaccination. The lesions have a similar localization but are not associated with oedema. The condition affects patients of both juvenile and adult ages and may recur [18].

ITP occurs mainly in younger children, as with AHEI, but can be diagnosed at any age. Petechiae, purpura and ecchymoses can be present in any location, although they have a mainly mucocutaneous predominance. Oedema is absent, bleeding is common, and the platelet count reveals significantly lower values.

Differentiation is also needed for purpura fulminans and sweet syndrome. Purpura fulminans is a rare syndrome of intravascular thrombosis and hemorrhagic infarction of the skin that is rapidly progressive and is accompanied by vascular collapse and disseminated intravascular coagulation. It is classified as a neonatal, idiopathic, or acute infectious disorder. Renal involvement is greater, with alterations in coagulation tests and sometimes acute renal failure [19].

Sweet’s syndrome (acute febrile neutrophilic dermatosis) is a hypersensitivity reaction that occurs in response to systemic factors, such as infection, hematologic disease, vaccination, inflammation or drug exposure. The condition is neutrophil-mediated. Typical skin lesions are bright-red, violet or reddish-blue papules, plaques, or nodules. Massive sub-epidermal oedema can produce a deceptively vesicular appearance, with pustules sometimes studding these lesions. The condition affects adults, involves both the trunk and the limbs and is associated with multiple autoimmune manifestations [20].

## 4. Conclusions

This case shows the classic skin lesions of AHEI that need to be differentiated from those of more severe diseases that require prompt recognition and therapy. In this case, the age of the patient, the lack of systemic involvement and the favorable clinical course without therapy were typical. However, as these patients may present to the emergency department with an impressive clinical picture, the condition must be diagnosed as quickly as possible to avoid unnecessary diagnostic procedures and to reassure parents.

## Figures and Tables

**Figure 1 ijerph-16-00823-f001:**
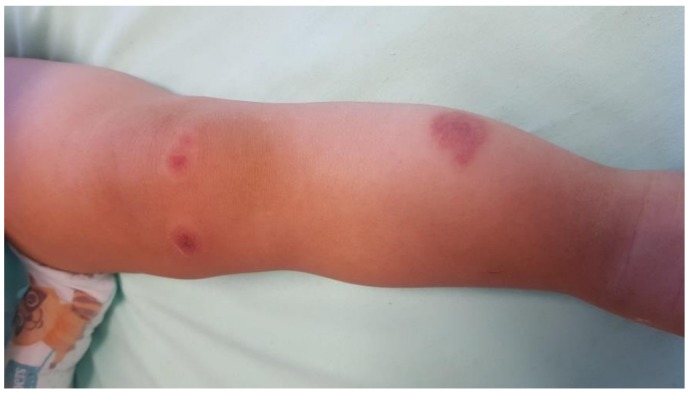
Petechial-hemorrhagic lesions with an oedematous component.

**Figure 2 ijerph-16-00823-f002:**
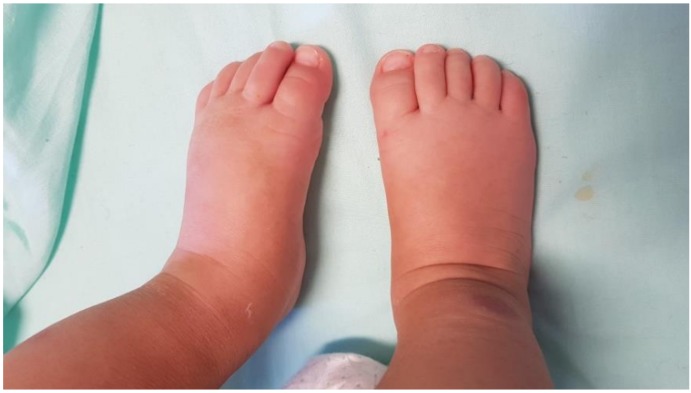
Oedema of the feet and ankles.

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
