# Peer review of "Targetoid Skin Lesions in a Child: Acute Hemorrhagic Oedema of Infancy and Its Differential Diagnosis"

_ijerph, 2019, doi:10.3390/ijerph16050823_

Round 1
Reviewer 1 Report
This article presents a case of AHEI in a 15-month old girl. The clinical presentation of disease is discussed, in addition to comparison with similar diseases. Overall, this paper is well-written. Reviewer suggests revisions to the following:
Line 67: Please provide the number of days post-presentation, in addition to the day post-hospitalization.
Line 86/87: Please provide a more recent reference for the number of annual cases. This reference is a decade old.
Line 92/93: Correlation to vaccination is referenced. Please provide additional details on the vaccine administered (is this the MMR vaccine as well?) and clarify the number of patients with this correlation (currently listed as 'some').
Line 104: Suggestion rewording for 'diagnosis was easy'.
Line 105/106: Although stated that drug treatment was not needed, please clarify if it is not needed because it alters the course of the disease or if they actually provide some level of relief to dermal symptoms.
Line 110: Suggestion revising 'important' with 'common', or similar.
Overall Discussion comment: The discussion is presented well, with details on a number of disease that may be confused with AHEI. However, this discussion does not clearly state why these other diseases discussed may be ruled out in direct comparison to general presentation of AHEI. Please add this clarification.
Overall Conclusions comment: The results and discussion do not clearly support the statement in the abstract and conclusions, "...the condition must be promptly diagnosed...". The presentation of AHEI in this case was not a prompt diagnosis, and, in the reviewer's opinion, further clarification of how differentiation between AHEI and other common diseases may be implemented is warranted prior to acceptance of this conclusion.
Author Response
This article presents a case of AHEI in a 15-month old girl. The clinical presentation of disease is discussed, in addition to comparison with similar diseases. Overall, this paper is well-written.
Re: Thank you very much for your suggestions. We revised the manuscript accordingly.
Reviewer suggests revisions to the following:
Line 67: Please provide the number of days post-presentation, in addition to the day post-hospitalization.
Re: The number of days post-presentation has been added.
Line 86/87: Please provide a more recent reference for the number of annual cases. This reference is a decade old.
Re: A recent reference with a new number of reported cases of AHEI has been included.
Line 92/93: Correlation to vaccination is referenced. Please provide additional details on the vaccine administered (is this the MMR vaccine as well?) and clarify the number of patients with this correlation (currently listed as 'some').
Re: Data on AHEI associated with vaccination have been included.
Line 104: Suggestion rewording for 'diagnosis was easy'.
Re: All the sentence has been rewritten avoiding “diagnosis was easy”.
Line 105/106: Although stated that drug treatment was not needed, please clarify if it is not needed because it alters the course of the disease or if they actually provide some level of relief to dermal symptoms.
Re: In the text, it is specified that drugs are not needed because they do not reduce clinical manifestations and do not alter disease course.
Line 110: Suggestion revising 'important' with 'common', or similar.
Re: “Important” was deleted and “common” included. However, to clarify the text all the sentence was rewritten.
Overall Discussion comment: The discussion is presented well, with details on a number of disease that may be confused with AHEI. However, this discussion does not clearly state why these other diseases discussed may be ruled out in direct comparison to general presentation of AHEI. Please add this clarification.
Re: It has been specified that AHEI must be differentiated from diseases with similar clinical manifestations because some of them need prompt therapy as they can have a very rapid negative evolution.
Overall Conclusions comment: The results and discussion do not clearly support the statement in the abstract and conclusions, "...the condition must be promptly diagnosed...". The presentation of AHEI in this case was not a prompt diagnosis, and, in the reviewer's opinion, further clarification of how differentiation between AHEI and other common diseases may be implemented is warranted prior to acceptance of this conclusion.
Re: It has been specified that diagnosis of AHEI and differentiation from diseases with similar signs and symptoms must be made as fast as possible to avoid unnecessary diagnostic procedures.
Reviewer 2 Report
The case reported in the manuscript is helpful for the AHEI diagnosis. But I have some concerns.
The research progress of AHEI should be described briefly in the Introduction section.
The general background of the patent could be provided, at least including race, skin color and hair color.
Author Response
The case reported in the manuscript is helpful for the AHEI diagnosis. But I have some concerns.
Re: Thank you very much for your comments. We revised the manuscript accordingly.
The research progress of AHEI should be described briefly in the Introduction section.
Re: In the Introduction, it has been reported that in recent years evaluation of described cases, including analysis of skin biopsies, has led to establish the characteristics of AHEI and its natural course.
The general background of the patent could be provided, at least including race, skin color and hair color.
Re: Suggested data regarding patient characteristics have been included.
Reviewer 3 Report
The manuscript of Miconi et al. represents a case study of acute haemorrhagic oedema of infancy (AHEI). A 15-month-old girl with negative personal medical clinical history was admitted for the sudden appearance of petechiae on the soft palate. The patient was moderately febrile during the following two days, with a maximum ear temperature of 38.3°C. The fever disappeared on the third day, whereas the haemorrhagic rash progressively increased and extended to the limbs, face and auricles associated with a strong oedematous component. Moreover, on the second day of hospitalization, bilateral oedema of the metacarpophalangeal joints with joint pain appeared. However, some questions still remain open.
Major revisions:
1. Do the authors suspect any connection between the measles-mumps-rubella vaccine and AHEI symptoms? Is this already known in literature?
2. Did the authors measure the size of purpura in diameter?
3. How long was increased C-reactive protein concentration measured?
4. Was AHEI with or without mucosal involvement?
5. Did the authors further examine mononuclear elements with hyperbasophile cytoplasm in the peripheral smear? Any presence of monocytes/macrophages (M1/M2). Could the authors show any H&E or other stainings? Histopathology is important to confirm the diagnosis as leukocytoclastic vasculitis is mediated by immune complexes. The vasculitis typically involves both capillaries and venules of the mid and upper dermis, with possible fibrinoid necrosis. The inflammatory infiltrate is predominately neutrophils with occasional eosinophils. Immunofluorescence studies may exhibit positivity for fibrinogen, C3, C1q, IgG, IgE, IgM, or IgA.
Author Response
The manuscript of Miconi et al. represents a case study of acute haemorrhagic oedema of infancy (AHEI). A 15-month-old girl with negative personal medical clinical history was admitted for the sudden appearance of petechiae on the soft palate. The patient was moderately febrile during the following two days, with a maximum ear temperature of 38.3°C. The fever disappeared on the third day, whereas the haemorrhagic rash progressively increased and extended to the limbs, face and auricles associated with a strong oedematous component. Moreover, on the second day of hospitalization, bilateral oedema of the metacarpophalangeal joints with joint pain appeared. However, some questions still remain open.
Re: Thank you very much for you comments. We revised the manuscript accordingly.
Major revisions:
1. Do the authors suspect any connection between the measles-mumps-rubella vaccine and AHEI symptoms? Is this already known in literature?
Re: Relationships between vaccines and AHEI have been detailed. It has been added the till now only two cases of AHEI associated with MMR vaccine has been reported.
2. Did the authors measure the size of purpura in diameter?
Re: Diameter of haemorrhagic lesions has been reported. However, skin lesion characteristics are also clearly evidenced by the photo.
3. How long was increased C-reactive protein concentration measured?
Re: C-reactive protein was no more measured, considering the prompt evolution of disease toward healing.
4. Was AHEI with or without mucosal involvement?
Re: It has been specified that initially lesions were evidenced on the soft palate. Mucosal involvement was documented
5. Did the authors further examine mononuclear elements with hyperbasophile cytoplasm in the peripheral smear? Any presence of monocytes/macrophages (M1/M2). Could the authors show any H&E or other stainings? Histopathology is important to confirm the diagnosis as leukocytoclastic vasculitis is mediated by immune complexes. The vasculitis typically involves both capillaries and venules of the mid and upper dermis, with possible fibrinoid necrosis. The inflammatory infiltrate is predominately neutrophils with occasional eosinophils. Immunofluorescence studies may exhibit positivity for fibrinogen, C3, C1q, IgG, IgE, IgM, or IgA.
Re: We agree that a skin biopsy would have better documented the case and strongly supported the diagnosis. However, as highlighted in the text skin biopsy was not performed to avoid an invasive procedure in a child with an already well-defined diagnosis and good general conditions.
Round 2
Reviewer 3 Report
References 2 and 3 need formatting.